# Gut Bacteria and Neuropsychiatric Disorders

**DOI:** 10.3390/microorganisms9122583

**Published:** 2021-12-14

**Authors:** Leon M. T. Dicks, Diron Hurn, Demi Hermanus

**Affiliations:** Department of Microbiology, Stellenbosch University, Private Bag X1, Matieland, Stellenbosch 7602, South Africa; 21076235@sun.ac.za (D.H.); 18613756@sun.ac.za (D.H.)

**Keywords:** gut microbiota, mental health

## Abstract

Bacteria in the gut microbiome plays an intrinsic part in immune activation, intestinal permeability, enteric reflex, and entero-endocrine signaling. Apart from physiological and structural changes brought about by gut bacteria on entero-epithelial cells and mucus layers, a vast number of signals generated in the gastro-intestinal tract (GIT) reaches the brain via the vagus nerve. Research on the gut–brain axis (GBA) has mostly been devoted to digestive functions and satiety. Less papers have been published on the role gut microbiota play in mood, cognitive behavior and neuropsychiatric disorders such as autism, depression and schizophrenia. Whether we will be able to fully decipher the connection between gut microbiota and mental health is debatable, especially since the gut microbiome is diverse, everchanging and highly responsive to external stimuli. Nevertheless, the more we discover about the gut microbiome and the more we learn about the GBA, the greater the chance of developing novel therapeutics, probiotics and psychobiotics to treat gastro-intestinal disorders such as inflammatory bowel disease (IBD) and irritable bowel syndrome (IBS), but also improve cognitive functions and prevent or treat mental disorders. In this review we focus on the influence gut bacteria and their metabolites have on neuropsychiatric disorders.

## 1. Introduction

The human gut hosts close to 4 trillion microorganisms and represents between 400 and 500 species [1,2]. Slightly outnumbered by our gut microbiota (1.3:1), it is no surprise that the genetic material they carry represents 99% of our total genetic makeup [2,3,4]. At natural birth, the gastro-intestinal tract (GIT) of an infant is largely colonized with microorganisms from the mother’s uterus and vagina [5,6]. However, bacteria from the placenta, amniotic fluid and circulatory system of the mother may reach the fetus before birth [7,8,9,10,11,12,13]. Mechanisms involved in the translocation of bacteria from the mother to the fetus have not been well documented. It may be that bacteria from the mother’s GIT are captured by dendritic cells (DCs) penetrating the gut epithelium and are then translocated to lymphoid tissue and the placenta [14,15]. This hypothesis was proven with the transfer of a genetically labelled strain of *Enterococcus faecium* from pregnant mice to off-spring [8,9]. Another study conducted on rats [16] have shown that bacteria can be transferred to off-spring during pregnancy and lactation. The presence of *Escherichia*, *Enterococcus*, *Staphylococcus* and *Propionibacterium* in murine blood isolated from the umbilical cord indicated that bacteria may reach the fetus via the placenta and bloodstream [9,13]. Dasanayake et al. [17] reported that *Actinomyces naselundii*, normally present in the oral cavity, may reach the uterus via the circulatory system. This was supported by high cell numbers of oral microbiota in the placenta of healthy mothers [11].

Drastic changes in maternal gut bacteria have been recorded throughout pregnancy. In 57% of pregnant women studied, cell numbers of proteobacteria and actinobacteria increased drastically [18]. The first three months of pregnancy is characterized by an increase in butyrate-producing *Faecalibacterium* and *Eubacterium* spp. During the last three months higher cell numbers of *Enterobacteriaceae, Streptococcus* spp. and proteobacteria have been reported. The latter is known to promote inflammatory responses, but is kept under control with elevated cytokine levels at the placental interface [19,20].

Infants are generally not affected by later changes in the mother’s gut microbiome and tend to maintain a bacterial population characteristic to that of the mother during the first three months of pregnancy. However, should the placenta of mothers be infected with *Prevotella* and *Gardnerella*, newborns may develop distinctive inflammatory responses [21,22]. Transfer of microorganisms to the fetus and colonization of the GIT is not only influenced by the mother’s health and changes in physiological conditions, but also by stress, alcohol, nicotine, and medication prescribed during pregnancy [21,22]. Detailed studies performed on the meconium of healthy fetuses, and the first stool of newborns revealed that *Streptococcus mitis*, *Lactobacillus plantarum*, *Escherichia coli*, *Klebsiella pneumoniae*, *Serratia marcescens*, staphylococci and enterococci are amongst the first bacteria to colonize the GIT [23,24,25].

Further development of the gut microbiome is highly dependent on the infant’s health. Unusually high cell numbers of *Bacteroidetes* have been isolated from the GIT of diabetic infants [26]. In another study [27], high cell numbers of lactic acid bacteria and enteric bacteria present in the meconium were associated with maternal eczema and respiratory problems later in life. Most researchers are of the opinion that vast changes in composition of gut microbiota occur during the first two years of an infant’s life [28]. A metagenomic study conducted on 98 infants and their mothers have shown that one-year-old infants delivered via Caesarean (C)-section hosted *Enterobacter hormaechei*, *Enterobacter cancerogenus, Haemophilus parainfluenzae*, *Haemophilus aegyptius*, *Haemophilus influenza*, *Haemophilus haemolyticus, Staphylococcus saprophyticus*, *Staphylococcus lugdunensis*, *Staphylococcus aureus*, *Streptococcus australis*, *Veillonella dispar*, *Veillonella parvula* and a few *Bacteroides* spp. [29]. In contrast, the GIT of same age infants vaginally delivered contained fewer species, with *Bacteroides*, *Bifidobacterium*, *Parabacteroides*, *Escherichia* and *Shigella* the core bacteria [29]. During the first 4 months of the screening program 52 MetaOTUs (metagenomic operational taxonomic units) identified in a group of mothers could not be in the GIT of their infants. The species were thus either not transmitted to the infants or did not colonize the GIT of infants during the first few months [29]. On the other hand, *Propionibacterium acnes*, *Streptococcus agalactiae* and *Veillonella* spp., identified in more than 10 newborns, were not detected in any of the mothers [29]. The developing of a gut microbiome is thus clearly far more complex than originally understood and the first 5 years seem to be the critical phase in developing a core group of microorganisms [30]. During these years, changes in gut microbiota are influenced by altering physiological conditions and diet. *Bacteroides* spp., for example, are associated with high-fat or high-protein diets and *Prevotella* spp. with high-carbohydrate diets [31].

Accumulating evidence concurs that abnormal or disturbed gut microbiota is a contributing factor to the pathophysiology of various neurological and psychiatric diseases, including anxiety and depression, major depressive disorder (MDD), schizophrenia, bipolar disorder, autism and obsessive-compulsive disorder (OCD). It is thus important to learn more about the effect a healthy, balanced, gut microbiome has on the CNS, but also understand the effect an imbalanced microbiome (GIT in dysbiosis) has on gut–brain communication.

Exploration of the intimate cross-talk between the gut and brain may further unveil novel approaches towards combatting various disorders associated with the GBA. This cross-talk extends across a multitude of pathways, involving endocrine, immune and neural mechanisms which depend on extensive interactions between gut microbes and host. It is thus important to explore signals produced by gut microbiota and study the influence these pathways have on neuropsychiatric disorders. This review addresses the influence of gut bacteria and their metabolites have on a select few neurological and psychiatric diseases.

## 2. Gut Microbiota Alters Neural Signals

The bidirectional communication between gut microbiota and the brain is illustrated in Figure 1. The first bacteria that colonize the GIT of a new-born are aerobic and convert lactose in breast or formula milk to organic acids and short chain fatty acids (SCFAs) [32]. The glucose component in milk is critical in the shaping of an infant’s gut microbiome [32], but also plays an important role in brain development [33,34]. This is especially true for vaginally born infants. As *Lactobacillus* spp. represents the largest component of vaginal bacteria [11], they may have a profound influence on the manifestation of the initial gut microbiome and may play a distinctive role in the development of the central nervous system (CNS).

Anaerobic bacteria in the large intestine produce acetate, lactate, butyrate and propionate. Butyrate acts as an inhibitor of histone deacetylase (HDAC) [12,35]. This is an important observation, as studies conducted on animals with HDAC inhibitors have shown promising results in the treatment of brain trauma and dementia [35]. Overproduction of HDAC has been implicated in neurological disorders such as Parkinson’s disease, schizophrenia, and depression [35]. On the other hand, an increase in acetylated histones (ACHs) elevates the expression of the *bdnf* (brain-derived neurotrophic factor) gene in the frontal cortex and hippocampus, stimulating brain development [36,37]. Decreased levels of BDNF are linked to mood changes, depression, and anxiety [38,39,40,41]. Studies on germ-free mice have shown lower levels of BDNF expression in the hippocampus [36,39]. Similar findings have been reported in mice treated with antibiotics and antimicrobial supplements [36,38]. Treatment of neurological disorders may thus vest in the control of SCFAs and HDAC levels. This emphasizes the importance of a well-balanced gut microbiome.

Fluctuation in butyrate levels may be due to inadequate numbers of intestinal butyrate-producing bacteria, or abnormal high binding of butyrate to free fatty acid receptors (FFARs) located on entero-epithelial cells (EECs) [35]. Butyrate also activates certain G-protein-coupled receptors (GPCRs) and is associated with multiple neurodegenerative disorders [12,35]. Butyrate is also known to promote regulatory T cells and subsequently produce inflammatory cytokines [42]. The increased anti-inflammatory response keeps *Proteobacteria* numbers in the GIT under control and, by doing so, also prevents the production of inflammatory cytokines [43]. Controlling butyrate levels in the GIT is important, as a decrease inhibits GPCRs and interrupts immune or endocrine responses [44]. Apart from this, butyrate and other SCFAs also modify the integrity of the blood–brain barrier (BBB), thus affecting the CNS and maturation of microglia [44,45]. In germ-free (GF) mice, the malfunctioning of microglia could be reversed by administering high levels of a combination of butyrate, propionate and acetate [46]. The function of acetate is different in that it crosses the blood–brain barrier and accumulates in the hypothalamus from where it controls appetite [47]. Activation of the hypothalamic-pituitary-adrenal (HPA) axis also affects the enteric nervous system (ENS) which, in turn, sends signals to EECs [12].

Butyrate induces the expression of the tight junction proteins claudin-2, occludin and cingulin [48]. This minimizes the translocation of microorganisms and their antigens across the gut wall and is described as an anti-inflammatory response [35]. Propionic acid displays properties similar to butyrate [35]. However, propionate may act as a neurotoxin and is associated with autism [49]. Translocation of bacteria and their antigens from the lumen to the circulatory system stimulate the secretion of pro-inflammatory cytokines such as interleukins (IL-6, IL-1b), tumor necrosis factor-alpha (TNF-α), and C-reactive protein [35,50,51]. Other studies have shown that an increase in these cytokines lead to changes in cognitive behavior and mood [48,52]. Immunologically induced GI barrier defects in rodents caused neurodevelopmental-related behavioral disorders [53]. Rodents exposed to specific pathogens showed anxiety-like behavior and impaired cognitive functions [13]. Obese mice on a high-fat diet produced offspring that were more prone to social and behavioral dysfunctions [45], confirming that gut microorganisms play a critical role in neural signaling and mental health. The role gut microorganisms play in control of behavior, mood, and stress-related brain disorders is a relatively young, but fast evolving, research field [36].

Given the substantial influence of the gut microbiota on neurodevelopment and sequential neurological health, a balanced gut microbiome is imperative for favorable brain development and a healthy mental status. This is especially important in neonates, as the brain is then most vulnerable to internal and external changes [13]. However, the brain is also susceptible to environmental and pathological adversities during adolescence and is thus sensitive to signals leading to neurodevelopmental and brain disorders. Growing up is associated with drastic changes in hormones. Although the composition of gut microbiota remains relatively stable during adulthood, changes in populations may still influence behavior [13]. The GIT secretes more than 20 hormones that bind to specific receptors that communicates with the CNS. Production of hormones is regulated by gut nutrient content, and the interaction between gut microbiota and intestinal epithelial cells [54,55]. Chemical signals generated by EECs, either directly or in response to microbial metabolites, travel through the ENS and regulates digestion, salivation, lacrimation, urination, defecation and sexual arousal [56]. A clear association exists between chronic stress and gut inflammation disorders, such as IBD and IBS [57]. Signals from the CNS are sent back to EECs and gut microbiota via the ENS and peripheric nervous system (PNS) [58]. In a healthy person, the bi-directional flow of information through the GBA helps to keep the gut microbiota in a homeostatic state.

## 3. Gut Microbiota Regulates Serotonin Levels

Serotonin (5-hydroxytryptamine or 5-HT) plays a vital role in neuronal and endocrine signaling pathways [53] and is involved in the regulation of appetite, sleeping patterns, mood, and cognition [40,48]. Although serotonin is synthesized by enterochromaffin cells (EC) and neurons of the ENS (Figure 2), more than 80% is produced in the GIT by *E. coli*, and species of *Lactococcus, Lactobacillus, Streptococcus, Morganella*, *Klebsiella, Hafnia, Bacteroides*, *Bifidobacterium*, *Propionibacterium*, *Eubacterium*, *Roseburia* and *Prevotella* [48,53]. Enteric serotonin levels are regulated by tryptophan hydroxylase TPH1 and serotonin from the ENS by tryptophan hydroxylase TPH2 [59]. Furthermore, the expression of *Tph1* (one of two tryptophan hydroxylases), is induced by SCFAs [53], whereafter TPH1 modulates EC-cell derived serotonin [59]. This confirms the association of elevated levels of SCFAs with a decrease in anxiety and depression-like behaviors [48]. At physiological concentrations, SCFAs have been noted to cause an eight- to ten-fold increase in serotonin production, at least in an in vitro colonic mucosal system [12]. Excess serotonin is transported across the cell membrane by a serotonin reuptake transporter (SERT) and intracellularly inactivated by monoamine oxidase (MAO) [59]. Homologs of eukaryotic monoamine transporters produced by bacteria therefore play an important role in the distribution of serotonin in the gut mucosa. The precursor of serotonin tryptophan (Trp), present in the mucosal layer, modulates intestinal permeability. Elevated levels of serotonin cause a decrease in the permeability of the gut wall [48]. Additionally, low levels of serotonin lead to a decrease in the expression of occludin, thus increasing gut wall permeability. The latter was reported in patients diagnosed with IBS [48].

## 4. Role of Gut Microbiota in Psychiatric Disorders

The link between gut microbiota and disorders such as anxiety, depression, schizophrenia, bipolar behavior, autism, and obsessive-compulsive disorder (OCD) has been clearly demonstrated (summarized in Figure 3). Changes in gut microbiota with each of these disorders are listed in Table 1.

### 4.1. Anxiety and Depression

Stress, anxiety, mental illnesses, and methods used in treatment have a profound effect on the gut microbiome; reviewed by Cryan et al. [80]. Anxiety and depression may have a more profound effect on an infant. Animals administered specific strains of bacteria displayed changes in behavior [38,81,82,83]. The human GIT may be host to 3000 bacterial species, as recently reported by the Human Microbiome and MetaHIT studies [84,85,86]. A chronic inflammatory state of the GIT may lead to increased responsiveness to stress and to development of major depressive disorder, MDD [87]. Treatment with antibiotics not only change the microbiome, but may have a lasting effect on the brain, spinal cord and the ENS [88,89]. This may occur without changes in immune response, as animal studies have shown changes in behavior with low levels of microbial infections had little effect on immune activation [90]. On the other hand, individuals suffering from autoimmune disorders and chronic inflammation often develop comorbid depression [91]. Treatment of these individuals with proinflammatory agents such as interferon-alpha (IFN-α) led to an increase in depression [92]. This coincides with studies that linked an increase in the secretion of proinflammatory cytokines with changes in depression [93,94,95]. In some cases, chronic inflammation led to the disruption of the blood–brain barrier (BBB), caused cellular and structural changes in the central nervous system, and induced the release of glutamate from microglia [96]. Although studies done on animals and macrophages have shown that some antidepressants have anti-inflammatory properties [96,97,98,99,100,101], this is not supported by all studies [102,103]. There is, however, evidence that levels of IL-1β in the hippocampus, and its effect on hippocampal neurogenesis, is reduced by certain antidepressants [104,105,106].

Anxiety often develops from a young age and can lead to other mental disorders such as depression [6,107]. Major depressive disorder (MDD) is world-wide the leading cause of disability and is characterized by irritability, loss of concentration, loss of appetite and sleep, and depressed moods [108,109]. Since depression is often associated with a deficiency in the functioning of serotonin and/or norepinephrine at specific synapses in the brain, most of the currently available antidepressants prevents the reuptake of these biogenic amines into nerve terminals [87]. However, many patients treated for MDD developed resistance to antidepressants, which led to studies investigating the relationship between gut microbiota and depression [110,111].

Naseribafrouei et al. [60] studied the microbiota of 37 depressed patients by comparing 16S rRNA sequences of fecal bacteria with those isolated from non-depressed patients. Based on this study, *Bacteroides* spp. were present at low cell numbers in depressed patients, although high cell numbers of *Alistipes* and *Oscillibacter* spp. were recorded. Similar findings were reported by Jiang et al. [61] when 46 depressed and 30 non-depressed patients between ages 18 and 40 were studied. In addition, high cell numbers of *Clostridium* and *Roseburia*, but lower numbers of *Prevotella*, and *Ruminococcus* were reported in depressed individuals. Aizawa et al. [62] reported an underrepresentation of *Bifidobacterium* and *Lactobacillus* spp. in depressed patients and Valles-Colomer et al. [63] linked a reduction of *Coprococcus* and *Dialister* to depression. Dysbiosis observed in the GIT of depressed individuals may cause IBD and in some cases accentuate depression [111]. Zanoli et al. [112] reported an association between depression, Crohn’s disease, and cardiovascular complications. Common symptoms associated with IBD is diarrhea, rectal bleeding, intermittent nausea and abdominal pain or tenderness [57]. Although the authors [112] did not associate changes in gut microbiota with any of these symptoms, it is likely that an imbalanced microbiome did play a role.

*Oscillobacter* spp. are known to produce valeric acid, a compound that closely resembles gamma-amino butyric acid (GABA) and binds to GABA(a) receptors [113]. Binding of GABA to GABA(a) and GABA(b) receptors block CNS signals, which alleviates anxiety and depression [113]. With the binding of valeric acid to these receptors, GABA binding is inhibited, and the CNS signals are no longer blocked, resulting in anxiety. Of interest is the lowering in *Lactobacillus* cell numbers in patients that suffer from anxiety and depression. Certain species of *Lactobacillus* are responsible for GABA secretion as well as the neurotransmitter acetylcholine [114,115]. It may thus well be that low cell numbers of these *Lactobacillus* spp. contribute to anxiety and depression.

*Alistipes* spp. is associated with chronic fatigue and IBD [116,117]. Inflammatory factors produced by *Alistipes* spp. could play a role in depression and anxiety. Of interest was the low numbers of *Prevotella* spp. reported in patients with anxiety and depression. This is a conflicting finding, as *Prevotella* spp. are often associated with pro-inflammatory characteristics [118].

Studies on rats have shown that oral administration of *Faecalibacterium prausnitzii* (ATCC 27766) relieved anxiety and depression, suggesting that the strain may have psychobiotic properties [119]. The increase in SCFA levels in the cecum, and elevated plasma IL-10 levels, accompanied with a reduction in corticosterone and IL-6 levels, may explain the anxiolytic and antidepressant properties observed [104]. Lukic et al. [120] have shown that *Ruminococcus flavefaciens* upregulated genes involved in mitochondrial oxidative phosphorylation, whilst downregulating genes involved in synaptic signaling and neurogenesis. The authors [120] also reported a reduction in serotonin and norepinephrine in the prefrontal cortex. Studies such as these and reports on probiotic bacteria that influence neurotransmission, neurogenesis, expression of neuropeptides and neuroinflammation [58], opens a new research field in psychobiotics [121], especially probiotics affecting the CNS. For more information on probiotics and the effect on the nervous system, the reader is referred to the review published by Cryan et al. [80].

### 4.2. Schizophrenia

Schizophrenia is defined as a mental disorder characterized by abnormal thinking, perceptual disturbances, impaired memory, slow mental processing, and sporadic emotional expression [121,122]. This disorder affects at least 20 million people throughout the world and is amongst the top 10 global causes of disability [123]. Symptoms differ vastly and reasons for developing schizophrenia is not fully understood [122,124], apart that it manifests at adolescence and remains with the individual throughout life [125]. At least one study reported a connection between schizophrenia and early childhood development [122]. As the gut microbiome is drastically altered during the first few years of life, certain microbiota may play a role in the developing of schizophrenia. A recent study [65] established a link between the salivary microbiome and gut microbiota associated with schizophrenia. Although their findings confirmed previous reports of the association between salivary microbiota and anxiety, depression, and autism spectrum disorder, ASD [126,127,128,129], much more detailed observations were made. The study involved 208 individuals diagnosed with symptoms of first-phase schizophrenia, psychosis (high risk schizophrenia) and no symptoms (classified as healthy). Concluded from this study [65], Firmicutes had a competitive advantage over Proteobacteria and may live in synergy with actinobacteria, fusobacteria, and Acidobacteria during early stages of schizophrenia. The dominance of Firmicutes over Proteobacteria has also been observed in the salivary microbiome of patients with primary Sjögren’s syndrome [130], an autoimmune disease involving chronic inflammation of the salivary and lacrimal glands. Findings of Qing et al. [65] also suggest a switch towards microbiota that produce branched-chain amino acids (BCAA) and lysine in individuals with early-phase schizophrenia. This may indicate an increase in *Staphylococcus* and *Megasphaera*, as both genera has been associated with increased BCAA and lysine production [131,132].

The microbiome of schizophrenic patients, deduced from oropharyngeal studies, is largely represented by *Firmicutes*, especially lactic acid bacteria and in particular *Lactobacillus gasseri.* whereas Bacteriodetes and *Acinetobacteria* were in the minority [67]. In contrast to previous studies, the presence of *Proteobacteria* did not differ significantly between schizophrenic and non-schizophrenic patients. Yolken et al. [68] reported on the presence of bacteriophage *Lactobacillus* phage phi adh in schizophrenic patients. This phage prevails in the lysogenic state within *L. gasseri*, confirming that this species may have strong links to schizophrenia.

Nguyen et al. [64] were the first to report the effect an altered gut microbiome may have on schizophrenic individuals. Significantly lower levels of *Proteobacteria*, *Haemophilus*, *Sutterella*, and *Clostridium* spp. were reported in patients 30 to 76 years old. Cell numbers of *Anaerococcus* spp. remained unchanged compared to healthy individuals. In a separate study conducted on patients between 18 and 65 years of age [66], high levels of *Proteobacteria*, *Succinivibrio*, *Collinsella*, *Clostridium* and *Klebsiella* spp., but low levels of *Blautia*, *Coprococcus*, and *Roseburia* spp. were reported. Contradictory findings reported in these two studies suggests that age plays a major role in the extent to which gut microbiota may change in patients with schizophrenia. *Proteobacteria* within the gut is the most unstable over time compared to the other three main phyla, especially when in a non-healthy state [133]. Lipopolysaccharides produced by *Proteobacteria* elicits the production of proinflammatory cytokines such as interferon-γ (IFN-γ), TNF-α, and interleukin-1β (IL-1β) [134]. This may cause intestinal inflammation and modification of tight junctions in the gut wall, leading to several intestinal diseases [135]. Given that a healthy human gut microbiome is seen to be relatively stable over time, Shin et al. [136] has proposed that fluctuations of *Proteobacteria* in the GIT could indicate microbial dysbiosis and could potentially be used as a diagnostic criterion [133,137].

The observation of a higher abundance of *Anaerococcus* and *Collinsella* in schizophrenic individuals is of interest, as species from these genera produce butyrate [69,70]. *Coprococcus* and *Rosburia* were less prevalent in schizophrenic individuals. Although these bacteria are also known butyrate producers, they are underrepresented in the gut compared to *Anaerococcus* or *Collinsella*. A similar observation was made with studies on bipolar and autistic patients. In these individuals, butyrate-producing *Faecalibacterium* spp. were present in low numbers [138]. Reasons for different reports on populations of butyrate-producing bacteria is unclear. It may be that population differences amongst these bacteria play an important role in the regulation of pro-inflammatory cytokines, which in turn influences certain psychiatric disorders.

Most *Haemophilus* spp. are regarded commensal, but some may cause meningitis. As psychiatric disorders are associated with inflammation, *Haemophilus* spp. was expected to be present at high cell numbers. However, the genus was less prominent in schizophrenic patients [139]. *Sutterella* spp., associated with reduced inflammation and low blood glucose levels [140], are also less prominent in schizophrenic patients. *Clostridium* and *Oscillospira* were in a lower abundance in schizophrenic and OCD patients, respectively [64,76,141,142]. This suggests that *Clostridium* may play a different role in schizophrenic and OCD patients than in patients suffering from anxiety, depression and autism.

*Bacteroides fragilis*, often isolated from schizophrenic patients, plays an important role in CD4^+^ T cell activation by producing zwitterionic polysaccharides (ZPS) that bind to peptide-binding sites on class II molecules of antigen-presenting cells. This stimulates T cells to produce anti-inflammatory IL-10, IL-2 and IL-12, thus playing a key role in host immune response [143]. Since *B. fragilis* is Gram-negative and contains a LPS capsule, the species may promote inflammation and be considered a pathogen [144,145]. Low numbers of *B. fragilis* observed in schizophrenic individuals suggests that they are most likely not pathogenic.

### 4.3. Bipolar Disorder

Bipolar disorder is similar to schizophrenia and depression, and is characterized as recurrent episodes of depression, along with cognitive, physical, and behavioral changes that, if severe enough, can lead to mania [146]. According to the World Health Organization (WHO), bipolar disorder affects approximately 60 million people worldwide and presents a high risk of suicide. Lithium is the choice of treatment, due to the drug’s anticonvulsant and antipsychotic characteristics. However, up to 50% of patients undergoing treatment still experience severe bipolar episodes [147].

Individuals suffering from bipolar disorder may experience an increase in gut wall permeability [51]. Coello et al. [71] reported a significantly higher abundance of *Flavonifractor* in bipolar individuals. The genus is known for its ability to cleave quercetin [148], a flavonoid with anti-oxidative and anti-inflammatory properties [149]. Changes in flavonoid levels could thus play a role in bipolar disorder.

A decrease in *Faecalibacterium* and an increase in *Actinobacteria* and Coriobacteriaceae was reported in bipolar patients [72]. In another study by Evans et al. [73], a decrease in *Faecalibacterium* and Ruminococcaceae was recorded in bipolar patients. The decrease in *Faecalibacterium* suggests a decline in anti-inflammatory reactions [150].

Actinobacteria consists of many different genera, some of which are pathogens. *Bifidobacterium* spp. with probiotic properties have been associated with the alleviation of IBD [74,75]. Cell numbers of *Bifidobacterium* spp. were, however, lower in schizophrenic patients and individuals suffering from anxiety, suggesting that the GIT could be inflamed. The role other actinobacteria play in bipolar disorder has been less researched.

*Coriobacteriaceae* play an important role in bile salt and steroid conversion, and the activation of polyphenols [151]. Species from this family may, however, become opportunistic pathogens, but this must be confirmed.

### 4.4. Autism

Autism is a disorder characterized by restricted or repetitive behavior as well as difficulties with communication and social interactions [152]. Symptoms may manifest in infants as young as one year [153,154]. Although autism is considered to have a genetic origin, environmental factors may lead to the development of a series of co-occurring medical conditions, including anxiety [155,156].

Reports that as many as 90% of individuals diagnosed with autism suffer from dysbiosis led researchers to study the role gut microbiota play in such cases [157,158,159]. Mccartney et al. [76] reported a significant increase of *Clostridium* spp. in autistic individuals, supporting previous findings [141,160]. In a more detailed study on the complete microbiome of autistic patients, Finegold et al. [77] indicated a significant increase in Bacteroidetes, *Acintobacterium* and *Proteobacterium* spp., but a decline in *Firmicutes* in autistic patients. High cell numbers of *Clostridium defense*, *Clostridium hathewayi* and *Clostridium orbiscindens* were recorded in autistic patients. *Faecalibacterium* and *Ruminococcus* spp. were less abundant, which is an important observation given the anti-inflammatory properties of these species. Most of the *Clostridium* spp. are, however, considered commensal with a key role in maintaining gut homeostasis [161]. They also induce colonic T regulatory cells [162]. *Roseburia* spp., well represented in autistic patients, produce butyrate that has anti-inflammatory properties. Compared to controls, individuals diagnosed with autism had lower cell numbers of *Ruminococcus*, a genus within *Clostridium* cluster XIVa. *Ruminococcus albus* degrades cellulose and produces acetate [78].

### 4.5. Obsessive-Compulsive Disorder (OCD)

OCD is a psychiatric disorder characterized by recurrent, intrusive thoughts or obsessions, and ritualistic compulsions [79]. This condition can have a lifetime prevalence in 2.3% of the population, with a predominance in men [163]. OCD was originally classified as an anxiety disorder, similar to autism, but has now been classified as an obsessive-compulsive spectrum disorder.

Few studies have been conducted on the microbiome of individuals with OCD. Experiments on mice showed a decrease in OCD when treated with *Lactobacillus rhamnosus.* In humans, similar findings were reported with the administration of *Lactobacillus helveticus* [164,165]. These observations led Turna et al. [79] to conduct a detailed study on the microbial diversity of the GIT of OCD patients. The authors reported low numbers of *Oscillospira*, *Odoribacter* and *Anaerostipes* spp. in OCD patients. In addition to this, an increase in systemic inflammation markers were noted. *Odoribacter* produces butyrate and is considered an anti-inflammatory species [166]. A decrease in *Odoribacter* in OCD patients could thus lead to an increase in inflammation, which may be the onset of OCD.

## 5. Trace Amines Influence Cognitive Functions, Anxiety and Depression

Trace amines are endogenous compounds comprising of *β*-phenylethyalmine, *p*-tyramine, tryptamine, *p*-octopamine, and some of their metabolites [167]. They are also abundant in food and are produced, and degraded, by intestinal microorganisms. Six functional isoforms of trace amine-associated receptors (TAARs) have been identified in humans, i.e., TAAR1, TAAR2, TAAR5, TAAR6, TAAR8, and TAAR9. Of these, TAAR1 is the most thoroughly studied and has both central and peripheral roles. In the CNS, TAAR1 acts as a regulator of dopaminergic, glutamatergic, and serotonergic neurotransmission and is a novel therapeutic target for schizophrenia, depression, and addiction. TAAR1 also regulates nutrient-induced hormone secretion and may be a therapeutic target for diabetes and obesity. TAAR1 may also regulate immune responses by regulating leukocyte differentiation and activation [167].

Decarboxylation of L-phenylalanine, L-tyrosine, and L-tryptophan by aromatic L-amino acid decarboxylase (AADC; EC 4.1.1.28) leads to formation of the trace amines β-phenylethylamine (PEA), *p*-tyramine (TYR) and tryptamine (TRP) [168]. *p*-octopamine (OCT) and p-synephrine are formed in the presence of dopamine-b-hydroxylase (EC 1.14.17.1) and phenylethanolamine-N-methyl transferase (PNMT; EC 2.1.1.28), respectively [169,170]. It is, however, noteworthy to mention that the K_m_ value of AADC is within the solubility of many precursor amino acids [171,172], which suggests that the synthesis of PEA, TYR and TRP may depend on the regulation of AADC [173,174], or specific variants of AADC [175]. An example of this is an exon 3-depleted variant of AADC expressed in neuronal and non-neuronal cells that lacks the ability to decarboxylate L-DOPA and L-5-hydroxytryptophan [176]. An AADC variant without exons 11–15 is expressed in non-neuronal tissue [175]. The enzymatic activity of this variant is not known. AADC variants with no clear enzymatic activity has also been detected in pancreatic b cells [177].

Production of PEA, TYR, and TRP by commensal gut microbiota is well documented [178,179,180]. Decarboxylation of precursor amino acids in the stomach [181] and entero-epithelial cells [182] play an important role in host-microbiota interactions. Decarboxylation of precursor amino acids also takes place in the glia, blood vessels [183], kidneys [184], liver [185], lungs [186] and pancreas [177]. In the brain AADC activity is regulated by dopamine, serotonin and glutamate. The activity of AADC may, however, also be affected by systemic lupus erythematosus [187,188]. Unlike dopamine, norepinephrine, epinephrine and serotonin, PEA, TYR and TRP are not stored and rapidly diffuse across membranes [78,79,189,190]. PEA diffuses across the blood–brain barrier [191] and TYR across intestinal epithelial cells [192]. Tyrosine is converted to l-3,4-dihydroxyphenylalanine (l-DOPA), a precursor of the catecholamines dopamine, norepinephrine (noradrenaline) and epinephrine (adrenaline). A deficiency in L-tyrosine may thus lead to anxiety and low mood [192]. Treatments that increase monoamine neurotransmitter receptor activation leads to a decrease in PEA and TYR synthesis. Likewise, treatments that decrease receptor activation results in an increase in PEA and TYR synthesis. Reports on changes in AADC activity are almost exclusively based on L-DOPA as substrate. Binding of PEA, TYR, TRP and OCT to TAAR1 in the brain regulates the release of neurotransmitters dopamine and serotonin [167].

Inhibition of the reuptake of monoamine neurotransmitters occurs when PEA and TYR concentrations exceed 10 mM [193,194], which is 100-fold higher than physiologic concentrations [195]. Similar indirect sympathomimetic responses to OCT have been reported [194,196]. N-methylated metabolites of PEA, TYR, N-methylphenylethylamine, N-methyltyramine and N-methyl metabolite of TRP N,N-dimethyltryptamine (DMT) are TAAR agonists [170]. Under- or over-expression of TAAR1 may lead to schizophrenia, depression and addiction [197]. TAAR1 is expressed in key areas in the brain where dopaminergic, serotonergic, and glutamatergic neurotransmission is modulated. These reactions also occur in the amygdala, hypothalamus, rhinal cortices, and subiculum [197]. TAAR1 may thus be a novel target for the developing of antipsychotic, mood-stabilizing, and antidepressant drugs. Some TAAR1 agonists exert incretin-like activity that leads to an increase in insulin secretion. Since TAAR1 releases the hormones peptide tyrosine-tyrosine (PYY) and glucono-like peptide 1 (GLP-1), TAAR1 antagonists may regulate obesity.

Anxiety and depression are controlled by “blocking” neurotransmission. During early life, gamma-amino butyric acid (GABA), produced by GABAergic neurons, serves as a neurotransmitter [198,199]. Later in life, when GABAergic neurons mature, glutamate is transferred between synaptic cells. Adhesion of GABA to GABA receptors (GABARs) on the postsynaptic surface de-activates ion channels involved in the transfer of Na^+^, K^+^, Ca^2+^ and Cl^−^ [200]. The inflow of positively charged ions into a cell excites GABA. Outflow of these ions leads to the inhibition of GABA formation. Three classes of GABARs have been described, i.e., GABAR_A_, GABAR_B_ and GABAR_C_. GABAR_B_ is a G protein–linked receptor (GPLR) that directs signals received from pheromones, hormones and neurotransmitters to signal transduction pathways [200,201,202]. Glycoproteins, 80-kDa in size and containing multiple transmembrane regions, act as transporters of GABA. At least six different GABA transporters are known. The levels of unbound GABA in the cleft are tightly regulated by reuptake into presynaptic nerve terminals and surrounding glial cells [203]. Under normal physiological conditions, the intracellular level of GABA exceeds extracellular levels by approximately 200. The uptake of GABA by nerve cells occurs when Na^+^ levels decrease. In the glia GABA is converted to glutamine, which is transferred back to the neuron [203]. Glutamine is then converted by glutaminase to glutamate, which re-enters the GABA shunt. *Lactobacillus rhamnosus* JB-1 altered the expression of GABARs in the brain, which resulted in the reduction of anxiety-like and depressive behavior [81].

Acetate, propionate, and butyrate interact with G-protein-coupled receptors 41 (GPR41) and 43 (GPR43) on the surface of EECs [204]. This, in turn, leads to the expression of the *pyy* gene encoding PYY. Most of PYY is released from L cells in the mucosa of the ileum and colon [39]. At elevated PYY levels a loss in appetite is experienced, which leads to a decrease in the rate of gastric emptying and the sensation of fullness [205]. Since PYY is present in the ileum at high levels, it is often referred to as an “ileal brake” [206]. At a state of satiety, water uptake increases, and electrolytes accumulate in the colon, leading to an increase in nutrient uptake. Smaller quantities of PYY (1–10%) is released in the esophagus, stomach, duodenum and jejunum [207]. Cleavage of the Tyr-Pro amino terminal residues of PYY_1-36_ by dipeptidyl peptidase IV (DPP-IV) produces more PYY_3-36_. [208]. During fasting, PYY_1-36_ levels are much higher compared to PYY_3-36._ The latter is released within 15 min of food intake, thus before the ingesta reaches the lower part of the small intestine and colon [207]. This suggests that the initial post-prandial release of PYY_3-36_ is controlled by the CNS. Highest PYY_3-36_ levels have been recorded in the colon after approximately 90 min of food intake [209]. Secretion of PYY, GLP-1 and cholecystokinin (CCK) send signals to the vagus nerve. The levels remain high for up to 6 h. A diet rich in lipids increases PYY_3-36_ production, whereas a diet rich in proteins delays the release of PYY_3-36_ by as much as 2 h after a meal. Bile acids interact with the G protein-coupled bile acid receptor (GPCR) TGR5 (also known as GPBAR 1) and farnesoid X receptors (FXR) on EECs. Binding of SCFAs and bile to these receptors stimulate the secretion of gut hormones such as PYY, GLP-1 and CCK.

A protein-rich diet stimulates the production of CCK. The hormone interacts with CCK-A receptors on acinar cells in the pancreas, CCK-B receptors in the brain and stomach and other CCK receptors distributed throughout the CNS [54]. This sends a signal to the small intestine to stop gastric emptying, thus mediating satiety. CCK also stimulates the pancreas to release enzymes involved in the digestion of lipids, proteins and carbohydrates [54]. CCK also interacts with calcineurin in the pancreas, which in turn activates the transcription factors NFAT 1–3 [210]. The latter stimulates hypertrophy and growth of the pancreas. The release of CCK is inhibited by somatostatin and pancreatic peptide. Trypsin, released by the pancreas, hydrolyses the CCK-releasing peptide and shuts down further secretion of CCK. The presence of CCK stimulates the contraction of the gall bladder to increase the secretion of bile into the duodenum [211]. CCK cannot cross the blood–brain barrier, but certain parts of the hypothalamus and brainstem are not protected by the barrier. Gastrin, a gastrointestinal hormone, binds to CCK_B_ receptors, which stimulates the release of gastric acid and the production of mucosa. Studies conducted on humans and rodents have shown that elevated CCK levels increases anxiety [54].

## 6. Conclusions

Gut microbiota has an adverse impact on our GBA and overall mental health. Chemicals secreted by these bacteria, such as GABA, in addition to other metabolites, play an important role in anti-inflammatory responses and help alleviate psychiatric symptoms stemming from inflammation. Treatment of schizophrenic and bipolar patients with probiotics alleviated symptoms associated with IBD, autistic children benefitted from probiotic treatment and OCD-like behavior could be controlled. The effect IBD has on depression, stress and anxiety requires in-depth studies. Our understanding of exactly how gut microorganisms control cognitive behavior, mood, and neuropsychiatric disorders remains limited. The deciphering of this complex, everchanging network between cells and neurons requires in-depth research by scientists from diverse disciplines. Although preclinical and clinical investigations have shown that treatment with probiotics may improve mood, extensive and carefully controlled clinical trials need to be performed to evaluate the effectiveness in treating mental disorders. Biomarkers need to be developed to identify differences in the gut microbiome of individuals suffering from psychological disorders. Interactions between drugs used in treatment and gut microbiota need to be studied in greater depth. Studies should include multi-omics of gut and oral microbiota to have a better understanding of the mutual interplay between phyla. The identification of changes in the gut microbiome associated with psychological disorders may provide valuable information in the choice of treatment.

## Figures and Tables

**Figure 1 microorganisms-09-02583-f001:**
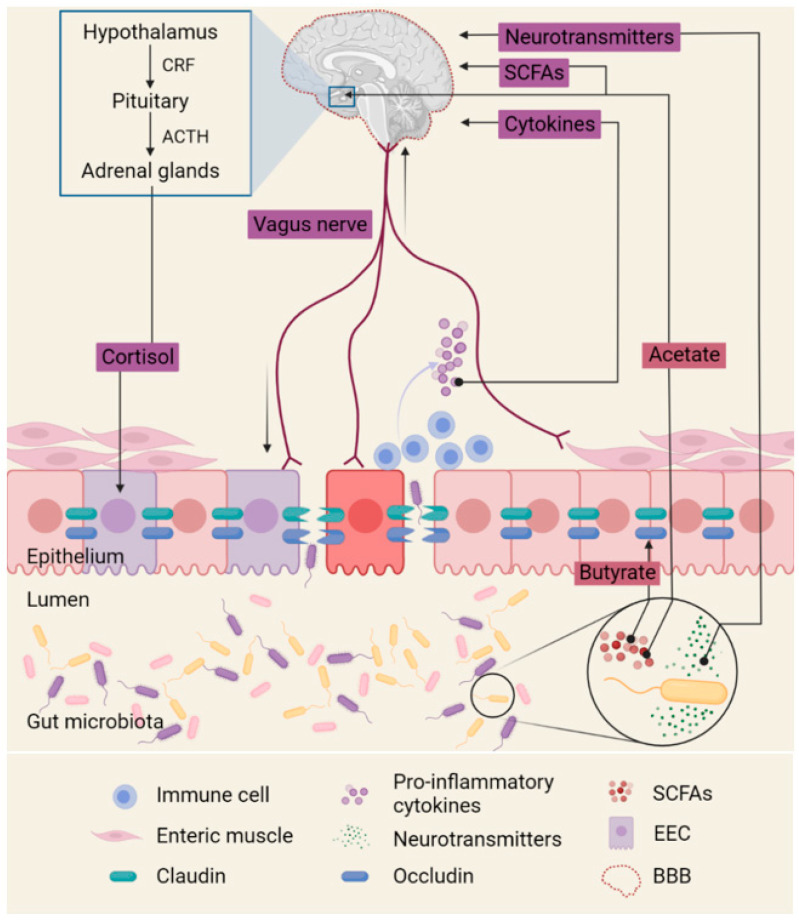
Mechanisms of bidirectional communication between gut microbiota and the brain. A network of entero-epithelial cells (EECs) along the gut wall mediates the bidirectional communication. In response to various stimuli and external cues, the central nervous system (CNS) modulate EECs via vagal efferents and the hypothalamic pituitary adrenal (HPA) axis. Gut microbiota return signals to the brain through different afferent pathways. Microbial metabolites, cytokine induction and neurotransmitters function via endocrine pathways; vagal afferents form part of the neurocrine pathway. Short chain fatty acids (SCFAs) produced by bacteria in the gut include acetate, lactate, butyrate and propionate. SCFAs modulate the integrity of the blood–brain barrier (BBB). Butyrate induces the expression of tight junction proteins, including claudins and occludins, and is therefore important for maintaining gut epithelial barrier integrity. A disrupted barrier encourages translocation of gut microbiota and their metabolites from the lumen to the circulatory system, resulting in the production of pro-inflammatory cytokines by immune cells, which can lead to changes in cognition and mood. Acetate crosses the BBB and accumulates in the hypothalamus, thereby controlling appetite. The bidirectional flow of information via the gut–brain axis can modify the gut microbiota and modulate behavior, mood and mental health.

**Figure 2 microorganisms-09-02583-f002:**
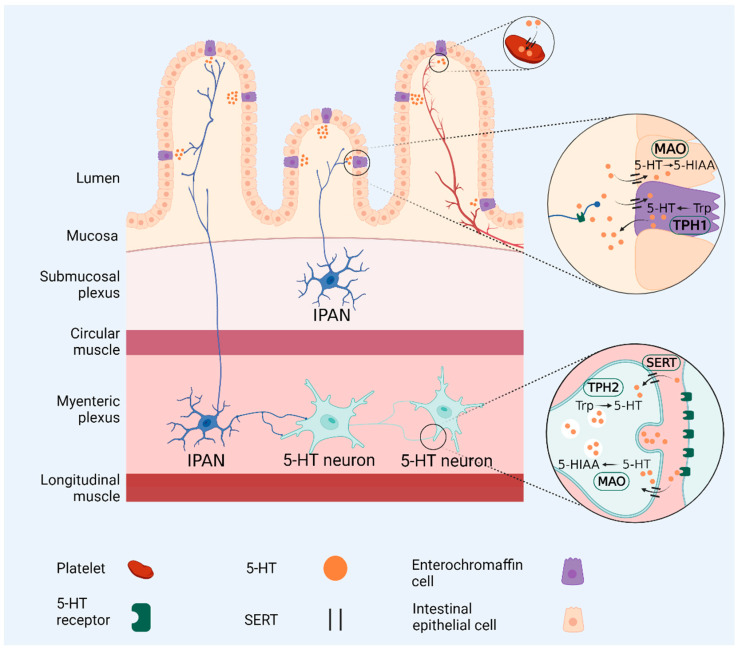
Synthesis, metabolism, and degradation of serotonin in the intestinal epithelium. Serotonin (5-hydroxytryptamine or 5-HT) is synthesized in the intestinal epithelium by enterochromaffin cells (ECs), serotonin-synthesizing neurons of the enteric nervous system (ENS) (5-HT neurons) and bacterial inhabitants of the gastro-intestinal tract (GIT). ECs convert tryptophan (Trp) to 5-HT with tryptophan hydroxylase 1 (TPH1). Enteric neurons use tryptophan hydroxylase 2 (TPH2) to convert Trp to 5-HT. Secreted 5-HT activates postsynaptic 5-HT receptors and is subsequently inactivated through pre-synaptic serotonin reuptake transporter (SERT) reuptake, where it can either be packaged into vesicles for release or degraded by monoamine oxidase (MAO). Release of 5-HT into the mucosal layer activates 5-HT receptors on intrinsic primary afferent neurons (IPANs) in both the submucosal and myenteric plexuses. SERT facilitates 5-HT inactivation. 5-HT is converted into 5-hydroxyindoleacetic acid (5-HIAA) by MAO. Platelets express SERT and are hypothesized to collect and store intestinal 5-HT as they move through enteric circulation.

**Figure 3 microorganisms-09-02583-f003:**
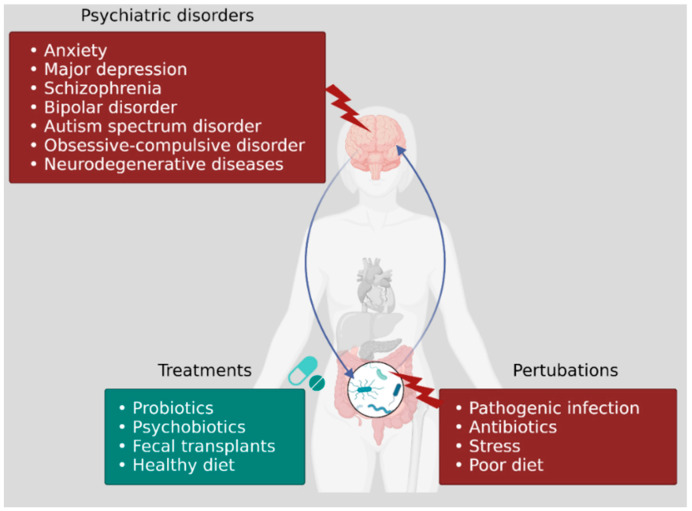
Link between perturbations in the gut microbiota and cognition, mood, and neuropsychiatric disorders. Disturbance of the gut microbiota may occur upon infection or administration of antibiotics, stress or with poor diet. There is evidence of a link between imbalances in gut microbiota and consequent psychiatric disorders, including anxiety and major depression, schizophrenia and autism spectrum disorder. Potential treatments include administration of probiotics to restore balance to the gut microbiota, fecal transplants from healthy individuals and maintaining a healthy, balanced diet.

**Table 1 microorganisms-09-02583-t001:** Changes in gut microbiota associated with mental disorders.

**Anxiety/Depression**	
**Reference**	**Findings**
[60,61]	↑ *Alistipes*, *Oscillibacter* ↓ Bacteroidales
[61]	↑ *Clostridium*, *Roseburia* ↓ *Bacteroides*, *Prevotella*, *Ruminococcus*
[62]	↓ *Bifidobacterium*, *Lactobacillus*
[63]	↓ *Coprococcus, Dialister*
**Schizophrenia**	
**Reference**	**Findings**
[64]	↑ *Anaerococcus* ↓ *Proteobacteria*, *Haemophilus*, *Sutterella*, *Clostridium*
[65]	↑ Firmicutes ↓ Proteobacteria ↑ Actinobacteria, Fusobacteria, Acidobacteria, *Staphylococcus, Megasphaera*
[66]	↑ Proteobacteria, *Succinivibrio*, *Collinsella*, *Clostridium*, *Klebsiella* ↓ *Blautia*, *Coprococcus*, *Roseburia*
[67]	↑ Firmicutes, *Lactobacillus gasseri* ↓ Bacteriodetes, Acinetobacteria
[68]	↑ *Lactobacillus* phage phi adh, *Lactobacillus gasseri*
[64]	↓ Proteobacteria, *Haemophilus, Sutterella, Clostridium*
[66]	↑ *Proteobacteria, Succinivibrio*, *Collinsella*, *Clostridium, Klebsiella* ↓ *Blautia*, *Coprococcus*, *Roseburia*
[69,70]	↑ *Anaerococcus, Collinsella*
**Bipolar disorder**	
**Reference**	**Findings**
[71]	↑ *Flavonifractor*
[72]	↑ *Actinobacteria*, Coriobacteriaceae ↓ *Faecalibacterium*
[73]	↓ *Faecalibacterium*, Ruminococcaceae
[74,75]	↓ *Bifidobacterium*
**Autism**	
**Reference**	**Findings**
[76]	↑ *Clostridium*
[77]	↑ Bacteroidetes, *Actinobacterium*, Proteobacteria, *Clostridium defense*, *Clostridium hathewayi, Clostridium orbiscindens* ↓ Firmicutes
[77,78]	↓ *Faecalibacterium, Ruminococcus*
[77]	↑ *Roseburia*
**OCD**	
**Reference**	**Findings**
[79]	↑ Systemic inflammation markers ↓ *Oscillospira*, *Odoribacter*, *Anaerostipes*

Arrows facing upwards (↑) denotes an increase in cell numbers and arrows facing downwards (↓) a decrease in cell numbers.

## Data Availability

Not applicable.

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
