# Peer review of "Gut Bacteria and Neuropsychiatric Disorders"

_microorganisms, 2021, doi:10.3390/microorganisms9122583_

Round 1
Reviewer 1 Report
This manuscript describes the latest evidence on the link between the gastro- intestinal tract microbiota and mental health. Although the subject is not new, the fact that new research and results are consistently produced in recent years, makes this review pertinent. A strong point in this scoping review is the fact that the biochemical pathways are well described and the connections between the microbial and host molecules is clearly established.
Figure 1 also provides a clear summary of the serotonin mediated interactions.
This comprehensive review seems complete, although recent studies have been published since the submission which may be useful and are listed at the bottom.
Another strong point of the manuscript submitted is the fact that it includes the "sampling" of the microbiome even before birth.
- doi: 10.1038/s41537-021-00180-1.
- doi: 10.3389/fneur.2021.721126.
- doi: 10.3389/phrs.2021.1603990.
Author Response
Answer: Information from these three papers have been added (see lines 262-285; 313 and 314; 343–347; 368-383).
- doi: 10.1038/s41537-021-00180-1.
- doi: 10.3389/fneur.2021.721126.
- doi: 10.3389/phrs.2021.1603990.
Reviewer 2 Report
The manuscript by Leon M.T. Dicks et al. summarized the effects of gut microorganisms on mental health. This topic is of interest to the readers. I have the following suggestions and comments:
1, mental health is a broad topic. However, the authors only discussed anxiety, depression, schizophrenia, bipolar disorder, autism and OCD. The authors should also discuss AD and PD. Otherwise the authors must revise the title.
2, More table should be added to summarize the changes of the gut microbiota and mental diseases.
3, The authors should discuss the future development in the field and should also give some suggestions. Also, what are the important questions in this field. This must be discussed.
4, Most of the discussed sections are just too preliminary. The authors must go behind the phenomenon to explore the molecular mechanisms. The molecular mechanisms linking gut microbiota and mental diseases must be discussed. The authors should add another figure to illustrate this point.
Author Response
The manuscript by Leon M.T. Dicks et al. summarized the effects of gut microorganisms on mental health. This topic is of interest to the readers. I have the following suggestions and comments:
1, mental health is a broad topic. However, the authors only discussed anxiety, depression, schizophrenia, bipolar disorder, autism and OCD. The authors should also discuss AD and PD. Otherwise the authors must revise the title.
Answer: AD and PD are neurological disorders, whilst our paper focuses on psychiatric disorders. We have changed the title of the paper to “Gut Bacteria and Neuropsychiatric Disorders”, which reflects the intention of the review much better.
2, More table should be added to summarize the changes of the gut microbiota and mental diseases.
Answer: The Table has been extended (changes are highlighted).
3, The authors should discuss the future development in the field and should also give some suggestions. Also, what are the important questions in this field. This must be discussed.
Answer: The Conclusion section has been amended to address these points.
4, Most of the discussed sections are just too preliminary. The authors must go behind the phenomenon to explore the molecular mechanisms. The molecular mechanisms linking gut microbiota and mental diseases must be discussed. The authors should add another figure to illustrate this point.
Answer: A figure has been added (Fig. 1) that, together with the other two figures, summarises the key points. Reference to Fig. 1 can be found in line 101.
Reviewer 3 Report
The manuscript entitled Gut Microorganisms and Mental Health is a very interesting work due to the increasing incidence of mental illness in recent times.
After reading it carefully, you will find my comments below.
Title: the authors speak only of bacteria, not of the complete microbiota profile. This should be specified in the title.
Likewise, a large part of the text focuses on the relationship of the chosen diseases with the metabolites of these bacteria. This should also be specified
If it is a review talking about the microbiota-mental illness relationship, in my opinion, it is essential to explain in detail the gut-brain and the reason for this connection. In this way, it will be possible to better understand why it is estimated that these diseases arise when dysbiosis is present.
The references should be updated, as most of the works mentioned are more than 5 years old.
The introduction is not an introduction/justification of the subject. Nor does it underline the objective of this review or its justification.
The part of the introduction that talks about the evolution of the composition of the microbiota since the individual is in the mother's womb should be a separate section.
Please, indicate the meaning of the abbreviation at the end of the paragraph. This will help the reader.
Figure 2, I would modify this figure by adding the gut-brain axis in more detail and the microorganisms and mechanisms named throughout the text. In a present way, it does not contribute anything new.
Table 1 (and the rest of the images/tables) check that all abbreviations are defined and omit references in the figure caption if applicable.
Define WHO.
Section 5, a short introduction to the section should be included, as the authors have been doing so far.
It is necessary to modify the conclusions section and discuss what has been mentioned and highlighted throughout this manuscript. The way it is now, by changing a couple of words, these "standard" conclusions could be applied to almost all review articles.
Author Response
The manuscript entitled Gut Microorganisms and Mental Health is a very interesting work due to the increasing incidence of mental illness in recent times.
After reading it carefully, you will find my comments below.
Title: the authors speak only of bacteria, not of the complete microbiota profile. This should be specified in the title.
Answer: The title has been changed to emphasize bacteria. New title: “Gut Bacteria and Neuropsychiatric Disorders”.
Likewise, a large part of the text focuses on the relationship of the chosen diseases with the metabolites of these bacteria. This should also be specified
Answer: This has now been addressed in the introduction (lines 84-98).
If it is a review talking about the microbiota-mental illness relationship, in my opinion, it is essential to explain in detail the gut-brain and the reason for this connection. In this way, it will be possible to better understand why it is estimated that these diseases arise when dysbiosis is present.
Answer: A figure has been added (Fig. 1) that, together with the other two figures, summarises the key points. Please also see the section added in introduction, specifically lines 91-98.
The references should be updated, as most of the works mentioned are more than 5 years old.
Answer: An additional nine to 12 references post 2019 have been mentioned and added to the reference list (also highlighted in the reference list). The re-numbering of references has been carefully cross-checked.
The introduction is not an introduction/justification of the subject. Nor does it underline the objective of this review or its justification.
Answer: The introduction has been amended (lines 84-98) to focus the reader’s attention on the aim of the review, i.e. the influence of gut bacteria and their metabolites on psychiatric disorders.
The part of the introduction that talks about the evolution of the composition of the microbiota since the individual is in the mother's womb should be a separate section.
Answer: We feel this section is important and correctly placed in introduction, and would like to keep this as is.
Please, indicate the meaning of the abbreviation at the end of the paragraph. This will help the reader.
Answer: We are not sure what the reviewer refers to, but have defined the abbreviation C-section (line 67).
Figure 2, I would modify this figure by adding the gut-brain axis in more detail and the microorganisms and mechanisms named throughout the text. In a present way, it does not contribute anything new.
Answer: We would like to keep this figure, as it summarises the association between psychiatric disorders, treatments and perturbations. The newly added figure (Fig. 1) provides additional information.
Table 1 (and the rest of the images/tables) check that all abbreviations are defined and omit references in the figure caption if applicable.
Answer: All abbreviations have been defined and references have been excluded from figure legends.
Define WHO.
Answer: Now defined (line 439).
Section 5, a short introduction to the section should be included, as the authors have been doing so far.
Answer: This has been added (lines 499-509).
It is necessary to modify the conclusions section and discuss what has been mentioned and highlighted throughout this manuscript. The way it is now, by changing a couple of words, these "standard" conclusions could be applied to almost all review articles.
Answer: The conclusion section has been amended.
Round 2
Reviewer 2 Report
The authors have addressed my major concerns. I suggest to accept this manuscript.
Reviewer 3 Report
The manuscript has improved from its previous version
Just a couple of comments
Figures: Define abbreviations at the end and not in the text of the caption.
Abbreviations are defined the first time they are used in the body of the text and then do not need to be defined further.